# Association of guideline and policy changes with incidence of lifestyle advice and treatment for uncomplicated mild hypertension in primary care: a longitudinal cohort study in the Clinical Practice Research Datalink

James P Sheppard,[1] Sarah Stevens,[1] Richard J Stevens,[1] Jonathan Mant,[2] Una Martin,[3] F.D. Richard Hobbs,[1] Richard J McManus[1]

[1]University of Oxford, Oxford, UK
[2]University of Cambridge, Cambridge, UK
[3]University of Birmingham, Birmingham, UK

**Correspondence to**
Dr James P Sheppard;
james.sheppard@phc.ox.ac.uk

## ABSTRACT

**Objectives** Evidence to support initiation of pharmacological treatment in patients with uncomplicated (low risk) mild hypertension is inconclusive. As such, clinical guidelines are contradictory and healthcare policy has changed regularly. The aim of this study was to determine the incidence of lifestyle advice and drug therapy in this population and whether secular trends were associated with policy changes.

**Design** Longitudinal cohort study.

**Setting** Primary care practices contributing to the Clinical Practice Research Datalink in England.

**Participants** Data were extracted from the linked electronic health records of patients aged 18–74 years, with stage 1 hypertension (blood pressure between 140/90 and 159/99 mm Hg), no cardiovascular disease (CVD) risk factors and no treatment, from 1998 to 2015. Patients exited if follow-up records became unavailable, they progressed to stage 2 hypertension, developed a CVD risk factor or received lifestyle advice/treatment.

**Primary outcome measures** The association between policy changes and incidence of lifestyle advice or treatment, examined using an interrupted time-series analysis.

**Results** A total of 108 843 patients were defined as having uncomplicated mild hypertension (mean age 51.9±12.9 years, 60.0% female). Patients spent a median 2.6 years (IQR 0.9–5.5) in the study, after which 12.2% (95% CI 12.0% to 12.4%) were given lifestyle advice, 29.9% (95% CI 29.7% to 30.2%) were prescribed medication and 19.4% (95% CI 19.2% to 19.6%) were given both. The introduction of the quality outcomes framework (QOF) and subsequent changes to QOF indicators were followed by significant increases in the incidence of lifestyle advice. Treatment prescriptions decreased slightly over time, but were not associated with policy changes.

**Conclusions** Despite secular trends that accord with UK guidance, many patients are still prescribed treatment for mild hypertension. Adequately powered studies are needed to determine if this is appropriate.

## Strengths and limitations of this study

► This study examined electronic health records from a large, nationally representative sample of the UK population.

► Analyses were undertaken according to a prespecified protocol which was reviewed and approved by an independent scientific advisory committee.

► Mild hypertension was defined consistently according to blood pressure readings, meaning that those with a diagnostic code for hypertension but no associated blood pressure values may have been excluded.

► Low-risk patients were defined according to individual risk factors, rather than an established risk calculator due to large amounts of missing data (eg, for cholesterol, body mass index), potentially underestimating the number of truly low-risk patients available for analysis.

► The incidence of lifestyle advice was defined according to clinical codes but it was not possible to identify patients given lifestyle advice where it was either not coded or where it was only recorded in the free text consultation notes.

## INTRODUCTION

High blood pressure (hypertension) is a key risk factor for the development of cardiovascular disease,[1] a major cause of mortality worldwide.[2] The threshold for hypertension is a sustained blood pressure >140/90 mm Hg, and many guidelines recommend pharmacological treatment is initiated immediately at this level, or after a period of unsuccessful lifestyle modification, regardless of the underlying risk of cardiovascular disease.[3–8] These recommendations are considered controversial, particularly with regard to treatment of uncomplicated (ie, low cardiovascular disease

risk) people with 'mild' hypertension (ie, sustained blood pressure between 140/90 and 159/99 mm Hg (stage 1)).[9-14]

There is little evidence to support initiation of pharmacological treatment in patients with uncomplicated mild hypertension. A Cochrane review[15] of trials in this population found no reduction in mortality or cardiovascular events with treatment, but was underpowered to detect significant differences.[10 15] More recent studies[16 17] have shown benefit with treatment in patients with mild hypertension. However, these included a significant proportion of patients with diabetes or at least one other cardiovascular risk factor and therefore participants would not be considered 'uncomplicated' under current clinical guidelines.[18]

Clinical guidelines often make recommendations based on 'expert opinion' where evidence is lacking or inconsistent. Such guidelines are predictably contradictory, with those in the UK promoting lifestyle modification in low-risk patients[18] while guidelines from Europe[6] and the USA[4 7 8] encourage prescription of drug therapy. Recently, an international expert consultation recommended early initiation of blood pressure-lowering treatment with lifestyle advice reserved only for patients with grade 1 isolated hypertension.[14] Against this background, a number of guideline and policy changes have occurred in the UK over the past 10 years which may have affected general practitioner's (GP's) treatment decisions. In particular, the introduction of a nationwide cardiovascular risk screening programme (the National Health Service (NHS) health check) in 2009,[19] and pay-for-performance indicators (the quality and outcomes framework (QOF)), which promoted both more lifestyle advice and stricter treatment targets in this population.[20]

The aim of this study was to use routine patient records from the Clinical Practice Research Datalink (CPRD)[21] to establish the proportion of patients given lifestyle advice and/or pharmacological treatment for uncomplicated mild hypertension in primary care and the association of policy with changes to these proportions over time.

## METHODS
### Design
This was a longitudinal cohort study in conducted in Primary Care from 1998 to 2015 using data from the CPRD, a database of electronic health records from England.[21] The incidence of lifestyle advice and treatment prescriptions was estimated by month, adjusted for age and sex. The association of guideline and policy changes was examined using an interrupted time-series analysis with introduction of the NHS health check[19] and changes in QOF indicators[20] defined as prespecified break points. The CPRD population has been shown previously to approximate to the UK population.[21]

### Study population
Individual patient data were extracted from the medical records of all patients registered at general practices contributing to the CPRD in England. Patients were entered into the study on the date of the third consecutive blood pressure reading between 140/90 and 159/99 mm Hg occurring after the study start date 1 January 1998. The start date was chosen because it represents the date from which all relevant data linkages were first available. Patients exited the study if they had three consecutive blood pressure readings >160/100 mm Hg (ie, progressed to stage 2 hypertension), developed a risk factor for cardiovascular disease, received lifestyle advice or treatment or follow-up records became unavailable (patient or practice left the CPRD) (table 1). The last day of follow-up for those remaining in the study was 30 September 2015 (last day of follow-up in linked data).

### Outcomes
The primary outcome of this study was evidence that lifestyle advice and/or pharmacological treatment had been given during follow-up. Lifestyle advice was defined as Read code for any lifestyle advice relating to smoking, alcohol, diet, hypertension, physical activity, weight, oral health or lipid disorders (see online supplementary appendix etable 1). Pharmacological treatment was defined as a coded prescription of an ACE inhibitor, angiotensin II receptor blocker, calcium channel blocker, thiazide or thiazide-like diuretic, beta-blocker, alpha-blocker or any other antihypertensive listed in the British National Formulary (see online supplementary appendix etable 2).

### Covariates
Data relating to baseline patient characteristics (age, sex, ethnicity, (practice level) deprivation), lifestyle factors (smoking status (never, current, ex-smoker), alcohol consumption (units per week), and body mass index (BMI)), pretreatment blood pressure readings (in the preceding 12 months), related comorbidities (rheumatoid arthritis, hypercholesterolaemia (Read code for hypercholesterolaemia or most recent total cholesterol value >7.5 mmol/L)) and all prescribed statin/antiplatelet medications were extracted from the CPRD database and used to define the study population.

### Quantitative data
Ethnicity was classified into white, black, South Asian, mixed race, other and unknown groups. The Index of Multiple Deprivation (IMD) was provided as quintiles of deprivation in England, with the highest quintile indicating the highest level of deprivation. Smoking status was defined as never, previous or current smoker. Those with unknown smoking status were assumed to have never smoked. Where there was no record of blood pressure lowering, statin or antiplatelet treatment, it was assumed that patients were not prescribed treatment.

**Table 1** Study entry and exit criteria applied to potential participant records used in the study

| Study entry criteria | Study exit (first to occur) |
|---|---|
| Date of the third consecutive blood pressure reading between 140/90 and 159/99 mm Hg (within 12 months of each other) | Last date at which the most recent linked data are available from the CPRD (study end date, September 2015) |
| Aged between 18 and 74 years | Date of third consecutive blood pressure reading >160/100 mm Hg (within 12 months of each other) (progression to stage 2 hypertension) |
| Linked general practice, Hospital Episodes Statistics and Office for National Statistics mortality records | Date of first record of a cardiovascular risk factor (left ventricular hypertrophy, atrial fibrillation, diabetes or chronic kidney disease or family history of premature heart disease) or high cardiovascular risk score (>20%) |
| Registered to a CPRD practice classified as 'up-to-standard' | Date of death |
| Classified as a CPRD 'acceptable patient' | Date of the most recent data upload from the practice to which a given patient is registered |
| No Read code for previous cardiovascular disease (stroke, myocardial infarction, angina, coronary heart disease, peripheral vascular disease, heart failure) | Date at which a given patient transfers out of a registered CPRD practice |
| No Read code for cardiovascular risk factor (left ventricular hypertrophy, atrial fibrillation, diabetes or chronic kidney disease or family history of premature heart disease) | Date of first coded lifestyle advice given after study entry (lifestyle advice analysis only) |
| No record of any blood pressure lowering medication prescription in the 12 months prior to the third consecutive blood pressure reading between 140/90 and 159/99 mm Hg | Date of first prescribed antihypertensive treatment given after study entry (treatment analysis only) |

CPRD, Clinical Practice Research Datalink.

Only patients with records deemed 'acceptable for research purposes' by CPRD were included in the analysis, and hence age and sex information was complete. Where there was no record of alcohol consumption, rheumatoid arthritis or hypercholesterolaemia, patients were assumed to be non-drinkers and have no history of these conditions. Where there was no record of blood pressure lowering, statin or antiplatelet treatment, it was assumed that patients were not exposed to any treatment. A total of 91 patients (0.001%) were missing IMD data and were excluded. To enable calculation of a cardiovascular risk score for sensitivity analyses, missing values for cholesterol and BMI were simulated once, assuming a normal distribution, and using means and SD in each age–sex strata from the 2011 Health Survey for England.[22]

### Sample size

As a descriptive study, no formal sample size calculation was required. All patients from the CPRD fulfilling the eligibility criteria were included.

### Statistical analysis

Descriptive statistics were used to describe the study population and estimate the proportion of patients being offered lifestyle advice or therapy (or both) following a diagnosis of uncomplicated mild hypertension.

The impact of the introduction of the NHS health check programme in April 2009[19] (month 111) and changes to QOF indicators (April 2009 and 2013 (month 159))[20] (see table 2) on lifestyle advice and treatment prescription rates were examined using an interrupted time-series analysis. A 2-year run-in phase (1998–1999)

was used, where patients could enter and leave the cohort, but incidence rates were not estimated. This was to ensure the cohort accumulated sufficient numbers of patients to enable accurate estimates of monthly incidence in the subsequent time period after the analysis start date (1 January 2000). The slope of the monthly rate of lifestyle advice and treatment over time was estimated (189 months in total), before policy change was introduced (months 111 and 159), along with the difference in the starting level (intercept) and slope of this rate in the period after each policy change was introduced, using ordinary least-squares regression with Newey-West corrected SEs (which account for autocorrelation). Estimates were adjusted for age and sex and all slopes are presented as incidence per year. P values indicate significance of the difference between the slope/intercept in the post-policy period, compared with estimates from the pre-policy period.

As a post hoc analysis, the impact of the introduction of the QOF in 2004[23] (month 59) was explored as an additional breakpoint in the interrupted time-series analysis. Sensitivity analyses explored the impact of excluding those deemed to be at high risk due to (1) high cardiovascular risk score documented in the primary care physician notes and (2) high cardiovascular risk score documented in the notes or estimated using the QRISK2 algorithm[24] and available risk factor data.

### Data access and cleaning methods

SS had access to the entire CPRD database from which eligible patients were identified and all data extracted.

**Table 2** Guideline and policy changes related to the ongoing management of hypertension

| Guideline/policy/QOF indicator | 1998/1999 | 1999/2000 | 2000/2001 | 2001/2002 | 2002/2003 | 2003/2004 | 2004/2005 | 2005/2006 | 2006/2007 | 2007/2008 | 2008/2009 | 2009/2010 | 2010/2011 | 2011/2012 | 2012/2013 | 2013/2014 | 2014/2015 | 2015/2016 |
|---|---|---|---|---|---|---|---|---|---|---|---|---|---|---|---|---|---|---|
| NICE Clinical Guideline 18 (2004): Essential hypertension: managing adult patients in primary care. | | | | | | | ✓ | | | | | | | | | | | |
| NICE Clinical Guideline 34 (2006): Management of hypertension in adults in primary care. | | | | | | | | | ✓ | ✓ | ✓ | ✓ | ✓ | | | | | |
| NICE Clinical Guideline 127 (2011): Hypertension in adults: diagnosis and management | | | | | | | | | | | | | | ✓ | ✓ | ✓ | ✓ | ✓ |
| Policy change: NHS health check cardiovascular risk screening programme introduced. | | | | | | | | | | | | | | | | ✓ | ✓ | ✓ |
| QOF hypertension indicator: The percentage of patients with hypertension in whom there is a record of the blood pressure in the previous 9 months. | | | | | | | ✓ | | ✓ | ✓ | ✓ | ✓ | ✓ | ✓ | ✓ | ✓ | ✓ | |
| QOF hypertension indicator: The percentage of patients with hypertension in whom the last blood pressure (measured in the previous 9months) is 150/90 or less. | | | | | | | ✓ | | ✓ | ✓ | ✓ | ✓ | ✓ | ✓ | | | | |
| QOF hypertension indicator: The percentage of patients aged 79 or under with hypertension in whom the last blood pressure reading (measured in the preceding 9 months) is 140/90mm Hg or less. | | | | | | | | | | | | | | | | ✓ | | |
| QOF hypertension indicator: The percentage of patients with hypertension who have not attained the age of 75 in whom there is an assessment of physical activity, using GPPAQ, in the preceding 12months. | | | | | | | | | | | | | | | | ✓ | | |
| QOF hypertension indicator: The percentage of patients with hypertension aged 16 or over and who have not attained the age of 75 who score 'less than active' on GPPAQ in the preceding 12months, who also have a record of a brief intervention. | | | | | | | | | | | | | | | | ✓ | | |
| QOF blood pressure indicator: The percentage of patients aged 40 or over who have a record of blood pressure in the preceding 5years. | | | | | | | | ✓ | ✓ | ✓ | ✓ | ✓ | ✓ | ✓ | ✓ | ✓ | ✓ | ✓ |

Continued

**Table 2** Continued

| Guideline/policy/QOF indicator | 1998/ 1999 | 1999/ 2000 | 2001/ 2002 | 2002/ 2003 | 2003/ 2004 | 2004/ 2005 | 2005/ 2006 | 2006/ 2007 | 2007/ 2008 | 2008/ 2009 | 2009/ 2010 | 2010/ 2011 | 2011/ 2012 | 2012/ 2013 | 2013/ 2014 | 2014/ 2015 | 2015/ 2016 |
|---|---|---|---|---|---|---|---|---|---|---|---|---|---|---|---|---|---|
| *QOF smoking indicator:* The percentage of patients with hypertension who smoke whose notes contain a record that smoking cessation advice or referral to a specialist service, where available, has been offered within the previous 15 months. | | | | | | ✓ | ✓ | ✓ | ✓ | ✓ | ✓ | ✓ | ✓ | ✓ | ✓ | ✓ | ✓ |
| *QOF smoking indicator:* The percentage of patients with hypertension whose notes record smoking status in the previous 15 months. Except those who have never smoked where smoking status need only be recorded once since diagnosis. | | | | | | ✓ | ✓ | ✓ | ✓ | ✓ | ✓ | ✓ | ✓ | ✓ | ✓ | ✓ | ✓ |
| *QOF smoking indicator:* The contractor supports patients who smoke in stopping smoking by a strategy which includes providing literature and offering appropriate therapy. | | | | | | | | | | | | | | | ✓ | ✓ | ✓ |
| *QOF primary prevention of cardiovascular disease indicator:* In those patients with a new diagnosis of hypertension: the percentage of patients who have had a face-to-face cardiovascular risk assessment at the outset of diagnosis using an agreed risk assessment tool. | | | | | | | | | | | ✓ | ✓ | ✓ | | | | |
| *QOF primary prevention of cardiovascular disease indicator:* The percentage of people diagnosed with hypertension who are given lifestyle advice for increasing physical activity, smoking cessation, safe alcohol consumption and healthy diet. | | | | | | | | | | | ✓ | ✓ | ✓ | ✓ | ✓ | | |
| *QOF primary prevention of cardiovascular disease indicator:* In those patients with a new diagnosis of hypertension, who have a recorded CVD risk assessment score of ≥20% in the preceding 12 months: the percentage who are currently treated with statins | | | | | | | | | | | | | | | ✓ | ✓ | ✓ |

Columns highlighted in dark grey indicate prespecified cut points, and column in light grey indicates additional post hoc cut point for interrupted time-series analysis.
CVD, cardiovascular disease; GPPAQ, General Practice Physical Activity Questionnaire; NHS, National Health Service; NICE, National Institute for Health and Care Excellence; QOF, quality and outcomes framework.

Where multiple blood pressure values appeared in the record on any one day, the lowest value was selected. Treatments/diagnoses of conditions were determined from the presence of at least one coded prescription/diagnostic code, respectively. Baseline data were extracted from within each patient's up to standard registration period and were drawn from the latest available record prior to study entry. Values deemed a priori to be clinically implausible were excluded (eg, weight values of <20 kg and >200 kg). Where there was evidence of prior smoking but the most recent smoking status recorded indicated no smoking, this was corrected by classifying patients as former smokers.

### Measures taken to reduce bias

To reduce the possibility of selection bias, all patients fulfilling the eligibility criteria for the study were included. Mild hypertension was defined according to measured blood pressure values rather than diagnostic codes, to reduce the potential for reporting bias, particularly in the pre-QOF era when coding was less standardised. For the primary analysis, breakpoints for the interrupted time-series analysis were prespecified based on policy changes thought likely to influence the giving of lifestyle advice and treatment, rather than trends observed in the data.

### Data linkage

Data linkage to the Office for National Statistics (ONS) mortality register was used to define patient mortality (patient exit from the study). Linkages to Basic Inpatient Hospital Episode Statistics (HES) were used in conjunction with CPRD records to define patient eligibility for the study (eg, previous stroke), and define patient ethnicity (where unavailable in primary care records). A linkage to the IMD was required to acquire practice level indices of multiple deprivation.

### Patient involvement

Patients were not involved in the development or design of this study.

### Approval

The present study protocol (16_008R) was approved by this committee in March 2016, prior to obtaining the data cut (protocol given in the online supplementary appendix). A project summary was published on the CPRD website (https://www.cprd.com/isac).

All data cleaning and analyses were conducted using STATA V.13.1 and 14.2 (MP parallel edition, StataCorp). Results are presented as means, medians or proportions, with SD, IQR or 95% CIs, unless otherwise stated.

## RESULTS

### Prevalence of lifestyle advice and treatment

Of the 7 416 968 individuals aged 18–74 years registered at linked, up-to-standard practices within the CPRD during the study period, a total of 108 843 patients (1.46%, 95% CI 1.45% to 1.47%) from 694 practices (333±192 patients per

practice) met the criteria for having uncomplicated mild hypertension during the study period (figure 1). Included patients had a mean age of 51.9±12.9 years, 60.0% (95% CI 59.7% to 60.3%) were female and mean blood pressure (across three visits) was 144.3±5.9/87.3±5.8 mm Hg (table 3). Patients spent a median time of 2.6 years in the study (IQR 0.9–5.5) and exited the cohort due to development of stage 2 hypertension (6269 patients (5.8%)), a cardiovascular risk factors or score >20% (3496 patients (5.0%)), advice being given (18 647 patients (17.1%)), treatment prescribed (35 305 patients (32.4%)), end of study follow-up (38 515 patients (35.4%)) or a combination of reasons (4656 patients (4.3%)). During the study period, a total of 13 269 (12.2%, 95% CI 12.0% to 12.4%) were given lifestyle advice alone, 32 578 (29.9%, 95% CI 29.7% to 30.2%) were only prescribed an antihypertensive and 21 128 (19.4%, 95% CI 19.2% to 19.6%) were given lifestyle advice and prescribed treatment (table 3). The most common type of lifestyle advice given was related to smoking (20.6%, 95% CI 20.4% to 20.9%; online supplementary appendix etable 3).

### Interrupted time-series analysis

In the pre-NHS health check period from January 2000 to April 2009, the proportion of patients receiving lifestyle advice increased from zero by 0.04% per year (95% CI 0.04% to 0.05%) (figure 2A). The introduction of the NHS health check and new QOF indicators in April 2009 was associated with a significant drop in lifestyle advice being given (p<0.001), but thereafter it increased by 0.19% per year (95% CI 0.15% to 0.23%). Following changes to QOF indicators in April 2013, the incidence of lifestyle advice decreased by −0.27% per year (95% CI −0.35% to −0.19%).

Treatment prescriptions were given to approximately 1.29% (95% CI 1.23% to 1.34%) of patients with uncomplicated mild hypertension in the first month of the study, but the incidence then decreased by −0.05% (95% CI −0.06% to −0.05%) per year in the pre-policy period until 2009 (figure 2B). Neither the introduction of the NHS health check, nor changes to QOF indicators significantly affected the incidence of treatment prescriptions which decreased by −0.03% per year and then −0.10% per year in the post NHS health check and QOF indicator change periods, respectively (figure 2B).

### Post hoc analysis

The introduction of the QOF and clinical guidelines for the management of hypertension in 2004 was associated with a step-change increase in the incidence of lifestyle advice of 0.15% (95% CI 0.05% to 0.26%; p=0.006) (figure 3A). Subsequently, there was a small but significant decrease in the incidence of lifestyle advice (−0.05% per year, 95% CI −0.07% to −0.04%). The NHS health check and new QOF indicators in 2009 were no longer associated with a significant drop in lifestyle advice when the introduction of the QOF (in 2004) was included as a cut-point in the post hoc analysis (p=0.07). No policy

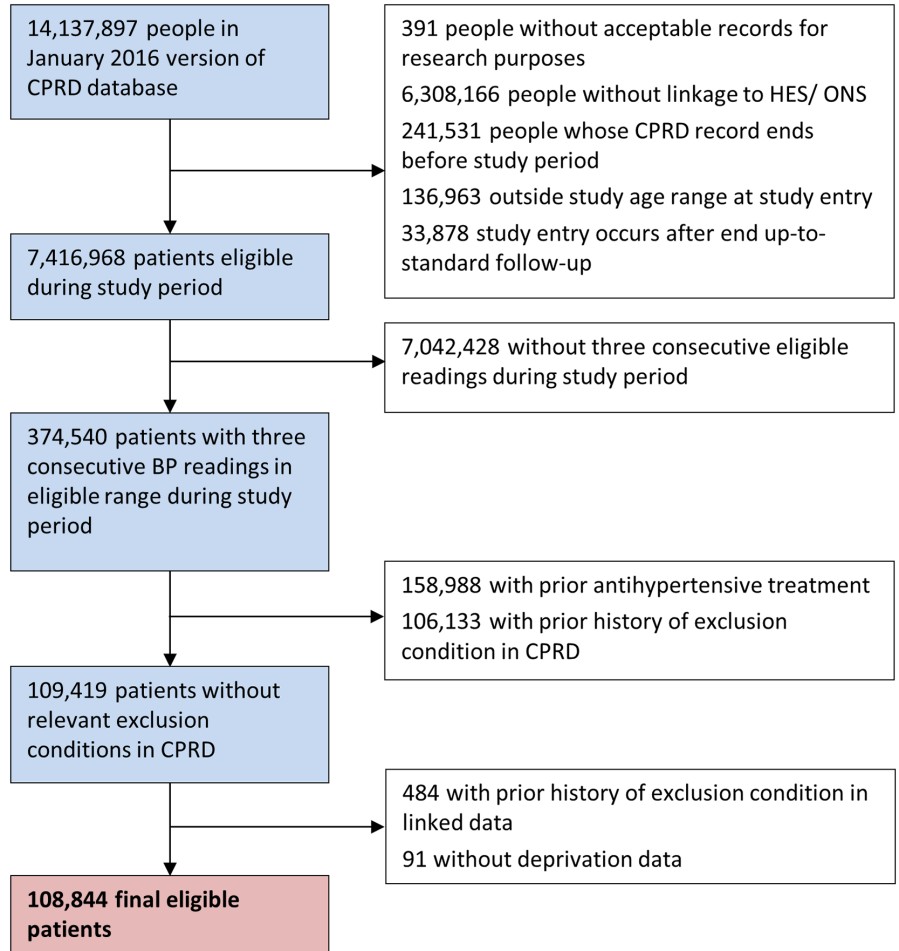

**Figure 1** Flow diagram showing exclusion of patient records and definition of the final cohort. BP, blood pressure; CPRD, Clinical Practice Research Datalink; HES, Hospital Episode Statistics; ONS, Office for National Statistics.

or guideline changes were associated with significant changes in the incidence of treatment prescriptions for uncomplicated mild hypertension during the study period in the post hoc analysis (figure 3B). Exclusion of high-risk patients (using either definition) in the sensitivity analysis had no impact on the incidence of lifestyle advice or treatment prescription.

## DISCUSSION

The present study examined a large database of electronic health records to establish the incidence of guideline recommended lifestyle advice and non-guideline recommended treatment for uncomplicated mild hypertension in primary care. Across the study period, approximately one-third of patients received lifestyle advice while half were prescribed antihypertensive treatment. The introduction of pay-for-performance indicators in 2004,[23] and more recently a cardiovascular risk screening programme,[19] was associated with an increase in the proportion of patients documented as having been offered lifestyle advice, suggesting a more guideline based approach to the management of uncomplicated mild hypertension. However, there remains an appreciable

proportion offered pharmacological treatment and this seems to be unaffected by changes to guidelines and policy.

### Strengths and limitations

This study examined electronic health records from a large, nationally representative sample of the UK population.[21] Participating practices included those which adhere to data quality standards set out by the CPRD to ensure accuracy of data studied. Analyses were undertaken according to a prespecified protocol (see online supplementary appendix) which was reviewed and approved by an independent scientific advisory committee, minimising the possibility of reporting bias and data mining.

In an attempt to define mild hypertension consistently across the population, it is possible that some patients with a diagnostic code for hypertension but no associated blood pressure values may have been excluded. The focus of this study was on 'uncomplicated' low-risk patients, but for pragmatic reasons, no attempt was made in the primary analysis to exclude those with a high risk of cardiovascular disease according to an established risk calculator[24 25] due to large amounts of missing data (eg, for cholesterol, BMI). Patients did exit the cohort if

**Table 3** Patient characteristics at study entry, stratified by subsequent outcome

| Characteristic | Total population | | Went on to receive no advice or treatment | | Went on to receive advice only | | Went on to receive treatment only | | Went on to receive advice and treatment | |
|---|---|---|---|---|---|---|---|---|---|---|
| | Mean/number | SD/% | Mean/number | SD/% | Mean/number | SD/% | Mean/number | SD/% | Mean/number | SD/% |
| Total population | 108843 | 100% | 41868 | 38.5% | 13269 | 12.2% | 32578 | 29.9 % | 21128 | 19.4% |
| Age (years) | 51.9 | 12.9 | 50.0 | 13.8 | 48.6 | 13.3 | 54.7 | 11.8 | 53.2 | 11.4 |
| Sex (% male) | 43567 | 40.0% | 15809 | 37.8% | 5443 | 41.0% | 13054 | 40.1% | 9261 | 43.8% |
| Time in study (years, median (IQR)) | 2.6 | 0.9–5.5 | 4.6 | 2.5–7.9 | 1.7 | 0.4–4.1 | 1.6 | 0.4–3.8 | 1.2 | 0.2–3.2 |
| Ethnicity | | | | | | | | | | |
| White ethnicity | 59643 | 54.8% | 21447 | 51.2% | 7554 | 56.9% | 17939 | 55.1% | 12703 | 60.1% |
| Black ethnicity | 1527 | 1.4% | 501 | 1.2% | 162 | 1.2% | 458 | 1.4% | 406 | 1.9% |
| South Asian ethnicity | 1356 | 1.2% | 437 | 1.0% | 133 | 1.0% | 487 | 1.5% | 299 | 1.4% |
| Mixed race ethnicity | 9700 | 8.9% | 3691 | 8.8% | 1483 | 11.2% | 2566 | 7.9% | 1960 | 9.3% |
| Other ethnicity | 1041 | 1.0% | 398 | 1.0% | 111 | 0.8% | 321 | 1.0% | 211 | 1.0% |
| Unknown ethnicity | 35576 | 32.7% | 15394 | 36.8% | 3826 | 28.8% | 10807 | 33.2% | 5549 | 26.3% |
| BMI (kg/m²)* | 29.3 | 5.8 | 29.2 | 5.9 | 29.4 | 6.2 | 29.2 | 5.6 | 29.3 | 5.7 |
| Current smoking status | 23592 | 21.7% | 9555 | 22.8% | 2286 | 17.2% | 7498 | 23.0% | 4253 | 20.1% |
| Alcohol (units per week)† | 12.1 | 14.6 | 11.2 | 12.5 | 15.9 | 19.7 | 10.6 | 13.0 | 13.5 | 16.0 |
| IMD score of 5 (most deprived) | 12606 | 11.6% | 4217 | 10.1% | 2081 | 15.7% | 3116 | 9.6% | 3192 | 15.1% |
| Systolic BP (mm Hg) | 144.3 | 5.9 | 143.7 | 5.9 | 143.3 | 6.1 | 145.2 | 5.6 | 145.0 | 5.7 |
| Diastolic BP (mm Hg) | 87.3 | 5.8 | 86.9 | 5.9 | 87.1 | 5.9 | 87.5 | 5.7 | 87.7 | 5.7 |
| Mean CVD risk score (mean±SD)‡ | 7.1% | 6.3% | 6.1% | 5.9% | 7.1% | 6.7% | 7.6% | 6.2% | 8.3% | 6.9% |
| Age-defined moderate CVD risk (%)¶ | 41510 | 38.1% | 14151 | 33.8% | 3887 | 29.3% | 14950 | 45.9% | 8522 | 40.3% |
| Statin prescription | 13426 | 12.3% | 3927 | 9.4% | 1472 | 11.1% | 4608 | 14.1% | 3419 | 16.2% |
| Antiplatelet prescription | 6029 | 5.5% | 1404 | 3.4% | 480 | 3.6% | 2567 | 7.9% | 1578 | 7.5% |

*Missing BMI data (n=46599).
†Missing alcohol data (n=72 785).
‡Based on previous risk score or QRISK2 score estimated from existing or imputed data.
¶Proportion of men aged ≥55 years and women aged ≥60 years who, according to Morales Salinas et al[14] should be classified as having moderate cardiovascular risk.
BMI, body mass index; BP, blood pressure; CVD, cardiovascular disease; IMD, Index of Multiple Deprivation.

**Figure 2** Primary analysis: incidence of lifestyle advice and treatment by month. (A) Lifestyle advice; (B) treatment Incidence estimates adjusted for age and sex. NHS, National Health Service; QOF, quality and outcomes framework.

they had a coded high cardiovascular disease risk score during follow-up. While it is possible that some high-risk patients may have been included in the study cohort initially, sensitivity analysis excluding patients with known high-risk scores and those with a high estimated QRISK2 score[24] did not affect the study results.

The incidence of lifestyle advice was defined according to clinical codes related to the giving of lifestyle advice which might be of relevance to the management of hypertension and cardiovascular disease risk (see online supplementary appendix etable 1). It was not possible to identify patients given lifestyle advice where it was either not coded or where it was only recorded in the free-text consultation notes. The trend of increasing lifestyle advice coinciding with the introduction of the QOF suggests that it may have been under-recorded, particularly in the early observational period.

### Previous literature

To our knowledge, this is the first study to determine the incidence of lifestyle advice and treatment in patients with uncomplicated mild hypertension. Previous studies[26–30] have focused on the prevalence of hypertension (defined as a blood pressure >140/90 mm Hg) and whether or not patients are treated and controlled, without making the important distinction between hypertensive patients where treatment is appropriate and those where it is not, at least according to national guidelines.[18] These surveys estimate the prevalence of hypertension across the world at between 20% and 45% with estimates of treatment and control ranging from 55%–81% and 30%–59%, respectively.[29] Our data suggest that in uncomplicated mild hypertension treatment rates are slightly lower (49% across the study period), perhaps reflecting uncertainty around its efficacy in this population.

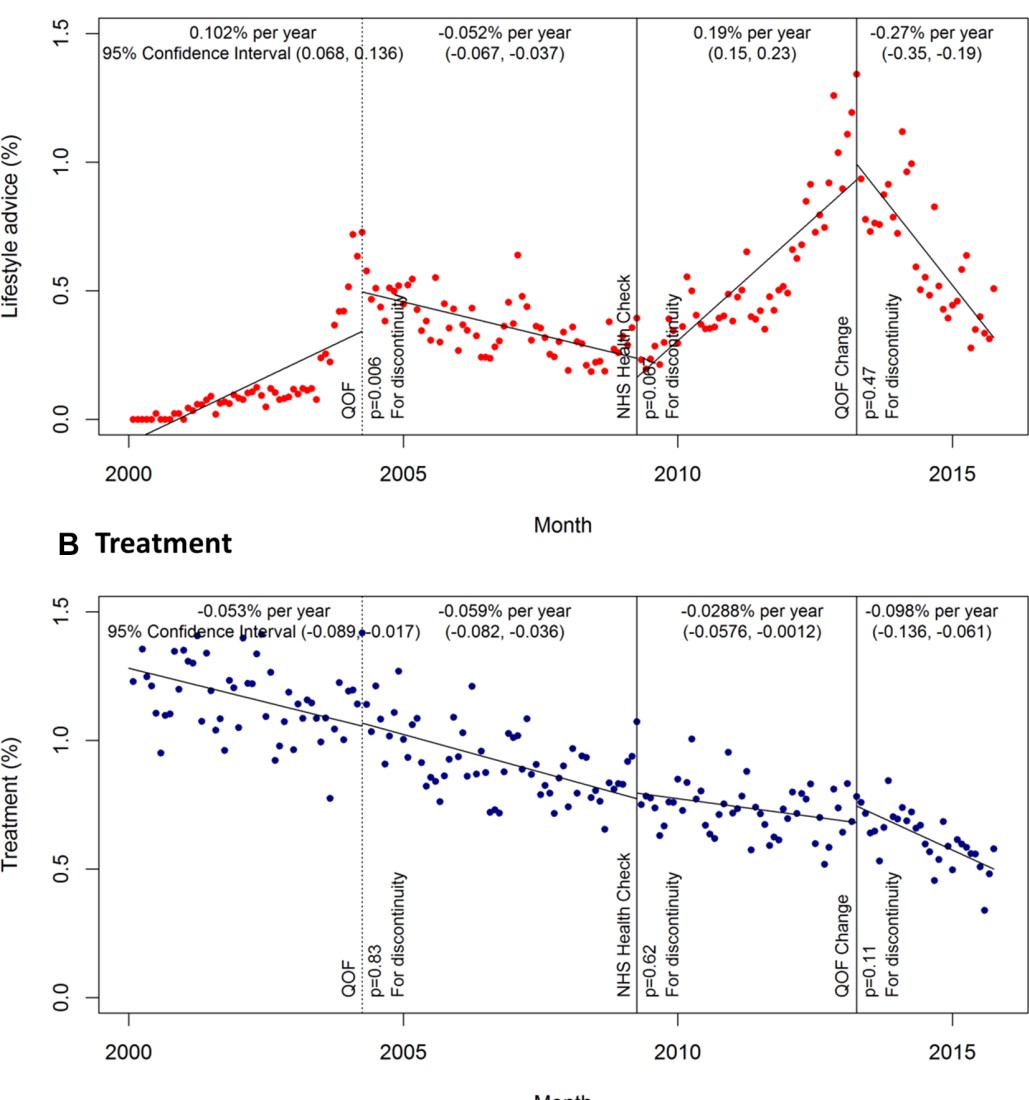

**Figure 3** Post hoc analysis: incidence of lifestyle advice and treatment by month. (A) Lifestyle advice; (B) treatment Incidence estimates adjusted for age and sex. NHS, National Health Service; QOF, quality and outcomes framework.

We found a trend towards increased recording of lifestyle advice over time, particularly following the introduction of the QOF in 2004,[23] the NHS health check[19] and changes in QOF indicators in 2009. Previous studies examining the impact of the NHS health check have found no effect on the estimated prevalence of hypertension,[31] but in those screened, referrals to specialist clinics and programmes for lifestyle advice increased sixfold.[32] The apparent association between incidence of lifestyle advice and changes in QOF indicators is in contrast to previous studies which have suggested that pay-for-performance in the UK has not affected the quality of care for hypertension.[33] While previous data have indicated that equivalent quality care remains, even after an indicator has been removed,[34] the present data suggest the opposite, with the incidence of recording of lifestyle advice decreasing after the indicator encouraging this was removed in 2014.[35]

The incidence of treatment prescriptions slightly decreased over time during the study with the largest reduction observed in the last study period (post 2013), perhaps due to more accurate blood pressure measurement and targeting of treatment brought about by new National Institute for Health and Care Excellence guidelines and the introduction of out-of-office blood pressure measurement for diagnosis in 2011.[18] The overall decrease in treatment might suggest a change in GP treatment practice over the past 15 years. However, it is also possible that GP treatment practice has remained constant over time, and the number of hypertensive patients being diagnosed and perceived as low risk has increased, as evidenced by the increasing prevalence of hypertension reported annually in the QOF over the past 10 years (from 12.5% to 13.8%).

## Implications for clinical practice

The peak in prevalence of lifestyle advice coinciding with the inclusion of QOF indicators encouraging advice for physical activity, smoking cessation and diet in 2013,[20] suggests that GPs were following pay-for-performance indicators closely. One explanation might be the content of QOF templates on GP computer systems, which are changed depending on the indicators included in the QOF and are likely to have encouraged better coding of lifestyle advice during a GP consultation. The choice of pay-for-performance indicators to include and exclude in the QOF should therefore be considered carefully, ensuring that such indicators reflect evidence-based practice and optimal patient care.

One could argue that the evidence for giving lifestyle advice in uncomplicated mild hypertension is as inconclusive as the evidence for pharmacological interventions, with many trials having short follow-up and exhibiting risk of bias[18 36] such as inadequate randomisation and allocation concealment,[36] lack of blinding[37] and incomplete outcome data.[38] In addition, there are no trials of the impact of lifestyle advice on 'hard outcomes' such as cardiovascular disease and mortality[18] and the quality of evidence of an association from observational studies is generally low.[39] However, lifestyle advice can potentially be given at lost cost within a patient consultation, and while overall reductions in blood pressure may be modest,[36–38] some patients may wish to change their lifestyle in an effort to delay or prevent embarking on a formal treatment regimen.[36]

## CONCLUSIONS

The present study shows that the introduction of pay-for-performance indicators and a cardiovascular risk screening programme have been associated with increased guideline recommended lifestyle advice being offered to patients with uncomplicated mild hypertension, although the overall prevalence remains relatively low. However, nearly half of the patients go on to be prescribed treatment, despite a lack of evidence suggesting benefit. This may be explained by a lack of clear evidence to guide treatment decisions. Adequately powered studies, perhaps using routine clinical data at scale, are required to accurately define the relationship between treatment and cardiovascular disease and death in patients with uncomplicated mild hypertension.

**Acknowledgements** The authors thank Dr Blanca Gallego Luxan for reviewing and commenting on the study protocol.

**Contributors** JPS conceived and designed the study, secured funding, undertook the main analysis and drafted the manuscript. SS was responsible for the data management, assisted with the analysis and revised the manuscript. RJS assisted with the study design, assisted with the analysis and revised the manuscript. JM assisted with the study design and revised the manuscript. UM assisted with the study design and revised the manuscript. RH assisted with the study design and revised the manuscript. RJM conceived the study, assisted with the study design and revised the manuscript.

**Funding** This work was funded by an Medical Research Council (MRC) Strategic Skills Post-doctoral Fellowship (MR/K022032/1) held by JPS. JPS now receives funding from the National Institute for Health Research (NIHR) Collaboration for Leadership in Applied Health Research and Care Oxford at Oxford Health NHS Foundation Trust and an NIHR Professorship which is held by and also funds RJMcM (NIHR-RP- R2-12-015). FDRH acknowledges support from the NIHR as Director of the NIHR SPCR, Director of the NIHR CLARHC Oxford, Theme Leader of the NIHR Oxford BRC, NIHR Oxford DEC and also from Harris Manchester College.

**Disclaimer** The views and opinions expressed are those of the authors and do not necessarily reflect those of the MRC, NHS, NIHR or the Department of Health.

**Competing interests** RJS is a member of the CPRD's Independent Scientific Advisory Committee (ISAC), but was not involved in the approval of this study.

**Patient consent** Not required.

**Ethics approval** The present study protocol (16_ 008 R) was approved by this committee in March 2016, prior to obtaining the data cut (protocol given in the online supplementary appendix). A project summary was published on the CPRD website (https://www.cprd.com/isac).

**Provenance and peer review** Not commissioned; externally peer reviewed.

**Data sharing statement** Data used in this analysis was provided by the Clinical Practice Research Datalink under a Gold Licence and cannot be shared with other parties.

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
