## [Reviewer comments · BMJ Open]

ARTICLE DETAILS

TITLE (PROVISIONAL)	Association of guideline and policy changes with incidence of lifestyle advice and treatment for uncomplicated mild hypertension in Primary Care: a longitudinal cohort study in the Clinical Practice Research Datalink
AUTHORS	Sheppard, James; Stevens, Sarah; Stevens, Richard; Mant, Jonathan; Martin, Una; Hobbs, Richard; McManus, Richard

VERSION 1 – REVIEW

REVIEWER	Guglielmo Trovato University of Catania, Italy
REVIEW RETURNED	20-Feb-2018

GENERAL COMMENTS	The manuscript Association of guideline and policy changes with incidence of lifestyle advice and treatment for uncomplicated mild hypertension in Primary Care: a longitudinal cohort study in the Clinical Practice Research Datalink is, in my view, overall interesting and timely; seemingly, further refinements may improve quality and interest of this article, particularly better displaying merits - by a scheme? - and limitations of the investigation performed It may contribute to advancement of science and health care delivery; if improved, it will read and possibly cited. An attempt for a more attractive and short title should be done. The individual participation of the Authors to the research should be better displayed
---

REVIEWER	Prof. Peter Schwandt MD Arteriosklerose Präventions Institut, München Germany
REVIEW RETURNED	20-Feb-2018

GENERAL COMMENTS	As long as a placebo-controlled 5-years follow-up study in more than 100,000 patients with mild hypertension and low risk for cardiovascular disease is unrealistic due to extreme costs we do not know whether these patients should be given medication to lower blood pressure. Therefore this study examined the extent to which patients are prescribed pharmacological treatment for uncomplicated, mild hypertension in routine practice and whether such treatment is safe and effective and reducing the risk of death and cardiovascular disease. Based on electronic health records the authors examined the incidence of lifestyle advice and drug therapy in a large longitudinal cohort study including 108,935 patients with uncomplicated mild hypertension from Primary Care practices. After a median 2.6 years 12% were given lifestyle advice, 30% were prescribed medication and 19% were given both. These results reflect real life conditions in adults defining low risk patients according to individually assessed
---

	risk factors, rather than using risk calculators. The most important result of this ambitious longitudinal study is that, nearly half of patients go on to receive treatment, despite a lack of evidence suggesting benefit. Treatment prescriptions decreased slightly over time, but were not associated with policy changes. Previous studies examining the impact of the NHS health check have found no effect on the estimated prevalence of hypertension, but referrals to specialist clinics and programmes for lifestyle advice increased did improve considerably 1. Significant increases in the incidence of lifestyle advice and treatment prescriptions decreased slightly over time after the implementation of the Quality Outcomes Framework (QOF) 2. The present study is a highly qualified additional approach for the prevention of non-communicable diseases without the unavoidable bias of scientific study designs which interfere with real life. 1. Because screening of cardiovascular risk factors in school children can predict risk of CVD risk factors in their parents³, data from young people in the data set could be of help implementing prevention in adults. 2. Furthermore, I would like to learn more about individual assessment of healthy lifestyle advice including nutrition and leisure time physical activity and its performance in this study. 3. Classifying patients with unknown smoking status as 'non-smokers' and those with unknown drug intake as 'not prescribed treatment' is misunderstanding because of the text on page 83 lines 44-46 "Where there is no record of blood pressure lowering, statin or antiplatelet treatment, it will be assumed the patients were not exposed to blood pressure lowering, statin or antiplatelet treatment". 4. P 81 line 58ff reads "Covariates relating to baseline patient characteristics will be extracted from the medical records of eligible patients". Some explanation would be helpful. 5. What do we learn from this study about the prescription of blood pressure lowering drugs in patients with coexisting mild hypertension and obesity with and without sustained control? Can this adult study be helpful for a comparable situation in young people 4? 6. Possibly I did not find informations like numbers of patients from how many primary prevention practices, comparability and quality control of devices used in all contributing practices in the manuscript. 1. Robson J, Dostal I, Sheikh A, et al. The NHS Health Check in England: an evaluation of the first 4 years. BMJ open 2016;6(1):e008840. doi: 10.1136/bmjopen-2015-008840 [published Online First: 2016/01/15] 2. NHS Employers GPC, NHS England. General Medical Services (GMS) contract Quality and Outcomes Framework (QOF): guidance for GMS contract 2014/15. Leeds: NHS Employers, 2014. 3. Schwandt P, Bischoff - Ferrari HA, Staehelin HB, Haas GM. Cardiovascular risk screening in school children predicts risk in parents. Atherosclerosis 2009; 250: 626 – 631 4. Schwandt P, Scholze JE, Bertsch T, Liepold E, Haas GM. Response to „Right analysis-wrong conclusion: obese youth with higher BP are at risk for target organ damage“. Am J Hypertension 2015; 28 (8):1072-1073
REVIEWER	Alberto Morales-Salinas

	Cardiocentro "Ernesto Che Guevara". Cuba
REVIEW RETURNED	28-Mar-2018

GENERAL COMMENTS	- The reviewer provided a marked copy with additional comments. Please contact the publisher for full details.
--

VERSION 1 – AUTHOR RESPONSE

Reviewer Comments to Author:

Reviewer: 1

1. The manuscript Association of guideline and policy changes with incidence of lifestyle advice and treatment for uncomplicated mild hypertension in Primary Care: a longitudinal cohort study in the Clinical Practice Research Datalink is, in my view, overall interesting and timely; seemingly, further refinements may improve quality and interest of this article, particularly better displaying merits - by a scheme? - and limitations of the investigation performed

Response:

We have revised the manuscript in line with the comments below and believe this has improved the quality of the article. The strengths and limitations are summarised after the abstract and in the discussion, as is recommended by BMJ Open. We have not amended the presentation of these, but are happy to be guided by the editors, if they feel changes to the presentation of the strengths and limitations are required.

2. It may contribute to advancement of science and health care delivery; if improved, it will read a and possibly cited. An attempt for a more attractive and short title should be done.

Response:

Our title describes the topic, sample population, study design and data source. These are requirements of the RECORD checklist and necessary to ensure adequate indexing in literature databases. We are not sure how this can be maintained in a shorter 'more attractive' title and therefore have not made any changes to the title.

3. The individual participation of the Authors to the research should be better displayed

Response:

The author contributions are clearly described in the section 'Author's contributions' on page 11, lines 389-395:

"JS conceived and designed the study, secured funding, undertook the main analysis and drafted the manuscript. SS was responsible for the data management, assisted with the analysis and revised the manuscript. RS assisted with the study design, assisted with the analysis and revised the manuscript. JM assisted with the study design and revised the manuscript. UM assisted with the study design and

revised the manuscript. RH assisted with the study design and revised the manuscript. RJM conceived the study, assisted with the study design and revised the manuscript”

Reviewer: 2

The present study is a highly qualified additional approach for the prevention of non-communicable diseases without the unavoidable bias of scientific study designs which interfere with real life.

1. Because screening of cardiovascular risk factors in school children can predict risk of CVD risk factors in their parents, data from young people in the data set could be of help implementing prevention in adults.

Response:

We thank the reviewer for this suggestion but note that the primary aim of this study was to examine the incidence of lifestyle advice and treatment prescriptions in patients recognised as having low risk mild hypertension in routine clinical practice. In the UK, screening of CVD risk factors in school children to predict their parents risk is very uncommon so including such variables in our cohort selection could identify a population which is not recognised as low risk by the treating physicians in the study. Such an approach could be explored in another study, but we have not made any changes to the present manuscript.

2. Furthermore, I would like to learn more about individual assessment of healthy lifestyle advice including nutrition and leisure time physical activity and its performance in this study.

Response:

We agree that this would be interesting but unfortunately detail of the advice is limited to the codes used by practitioners which are given in the appendix. We have now added the following table describing the frequency of different types of coded advice given during follow-up.

eTable 1. Frequency of different types of lifestyle advice given during follow-up

Advice type	Number of patients	%
Any lifestyle advice	34,397	31.6%
Advice about smoking	22,456	20.6%
Advice about diet	4,343	4.0%
Advice about exercise	3,778	3.5%
Advice about alcohol consumption	2,039	1.9%
Advice about weight	1,309	1.2%
Other advice	10,200	9.4%

Note: Some patients received more than one different type of advice

3. Classifying patients with unknown smoking status as 'non-smokers' and those with unknown drug intake as 'not prescribed treatment' is misunderstanding because of the text on page 83 lines 44-46 "Where there is no record of blood pressure lowering, statin or antiplatelet treatment, it will be assumed the patients were not exposed to blood pressure lowering, statin or antiplatelet treatment".

Response:

We have added the text from the original protocol to the methods to clarify this:

Page 5, lines 152-153

"Where there was no record of blood pressure lowering, statin or antiplatelet treatment, it was assumed that patients were not exposed to any treatment."

4. P 81 line 58ff reads "Covariates relating to baseline patient characteristics will be extracted from the medical records of eligible patients". Some explanation would be helpful.

Response:

This sentence comes from the pre-specified protocol in the appendix so it is not possible to make changes to this document. It simply means that data about eligible patients will be extracted from the CPRD database, which is a database of individual patient medical records from across the UK. We have clarified this in the methods section of the main paper:

Page 5 lines 133-139

"Data relating to baseline patient characteristics (age, sex, ethnicity, [patient level] deprivation), lifestyle factors (smoking status [never, current, ex-smoker], alcohol consumption [units per week], and BMI), pre-treatment blood pressure readings (in the preceding 12 months), related co-morbidities (rheumatoid arthritis, hypercholesterolemia [Read code for hypercholesterolemia or most recent total cholesterol value >7.5mmol/l]) and all prescribed statin/antiplatelet medications were extracted from the CPRD database and used to define the study population."

5. What do we learn from this study about the prescription of blood pressure lowering drugs in patients with coexisting mild hypertension and obesity with and without sustained control? Can this adult study be helpful for a comparable situation in young people?

Response:

These are interesting questions but not ones that can really be answered by the present data and analyses. We are keen to restrict the discussion to interpreting the pre-specified analyses which show that many individuals receive pharmacological treatment for uncomplicated mild hypertension which may not be evidence based and is unaffected by changes to clinical guidelines and policy. Stratification by obesity, or extension to young people, would be a *post hoc* analysis beyond the scope of the protocol. This new analysis could be explored in future work.

6. Possibly I did not find informations like numbers of patients from how many primary prevention practices, comparability and quality control of devices used in all contributing practices in the manuscript.

Response:

The number of participating practices is given in the results (page 7, line 233). We have now added the average number of patients per practice (333 ± 192 patients per practice; page 7, line 234). Unfortunately it was not possible to describe or assess the quality of devices used in practices since this is not captured in the electronic health records studied. Such limitations of EHR database studies must be offset against advantages, such as those this Reviewer acknowledged in his introductory remarks.

Reviewer: 3

1. Abstract: Proportions of patients exiting the study must be shown

We thank the reviewer for this suggestion. We agree this is worth reporting but this is difficult in the limited space available in the abstract. We have therefore added the following sentence to the results section instead:

Page 7, line 236-240

“Patients spent a median time of 2.6 years in the study (IQR 0.9, 5.5) and exited the cohort due to development of stage 2 hypertension (6,269 patients [5.8%]), a cardiovascular risk factors or score >20% (3,496 patients [5.0%]), advice being given (18,647 patients [17.1%]), treatment prescribed (35,305 patients [32.4%]), end of study follow-up (38,515 patients [35.4%]) or a combination of reasons (4,656 patients [4.3%]).”

2. Strengths and weaknesses: 2 more weaknesses a) study did not evaluate medication adherence and b) patients exited

Response:

We may have misunderstood the reviewer’s comments here, but we do not believe either of these represent weakness of the study:

a) The focus of this study was on practitioner behaviour – do they give lifestyle advice or prescribe therapy to patients presenting with uncomplicated mild hypertension? Since we examined incidence, those receiving a prescription of antihypertensive therapy subsequently exited the study and so it was neither possible nor relevant to examine consequent medication adherence. We take the point about adherence, which cannot currently be evaluated in this database. In most of the text, including the Results section, we believe it is already clear that our results pertain to “prescribed treatment” rather than received treatment, but in the Discussion we have made two changes. We have changed “receiving pharmacological treatment” in the first paragraph of the Discussion to “offered pharmacological treatment”. We have changed “half of patients go on to receive treatment” in the final paragraph of the Discussion to “half of patients go on to be prescribed treatment”.

b) In all observational cohort studies, patients must enter and exit the study. We specified exit criteria such as progression to stage 2 hypertension or development of CVD risk factors to ensure the cohort

under study remained representative of the intended population (i.e. uncomplicated mild hypertensives). We would argue that this is a methodological strength rather than a limitation. If those exiting the study at the end of follow-up were in some way different to those remaining for longer periods (i.e. some kind of informative censoring existed) then this might be considered a limitation of the study, however, we cannot think of a plausible rationale for why this would be the case. We have now quantified the number of patients who exited (see comment above).

3. European Guidelines only recommend treatment following a period of repeatedly raised BPs (in the mild hypertensive range) and following an unsuccessful trial of lifestyle modification

Response:

We thank the reviewer for pointing this out, and have clarified this in the introduction:

Page 4, lines 70-73

“The threshold for hypertension is a sustained blood pressure above 140/90mmHg, and many guidelines recommend pharmacological treatment is initiated immediately at this level, or after a period of unsuccessful lifestyle modification, regardless of the underlying risk of cardiovascular disease.”

4. Add reference: Morales Salinas, et al., (2017) Clinical perspective on antihypertensive drug treatment in adults with grade 1 hypertension and low-to-moderate cardiovascular risk: an international expert consultation. Current Problems in Cardiology, 42 7: 198-225.
doi:10.1016/j.cpcardiol.2017.03.001#

Response:

This has been added (page 4, line 76).

5. Add the following phrase: “Recently, an international expert consultation recommended an early initiation of BP lowering pharmacotherapy and have lifestyle advice reserved only for patients with grade 1 isolated hypertension” ref Current Problems in Cardiology, 42 7: 198-225.
doi:10.1016/j.cpcardiol.2017.03.001

Response:

This has been added (page 4, lines 89-91).

6. I would like to highlight that in the real world, hypercholesterolemia (total cholesterol >7.5) is a risk factor for CVD. I would like to know is cholesterol under 7.5 was considered a non CVD co-morbidity?

Response:

In the UK, hypercholesterolemia alone would not necessarily lead to antihypertensive treatment, unless it was also accompanied by a high CVD risk score. Practice may differ in other parts of the world, but it was decided that definition of risk in this cohort should match how it is defined in UK guidelines to which the consulting physicians would refer. We have now modified how our reference to hypercholesterolemia is phrased in the methods:

Page 5, lines 134-138

“Data relating to baseline patient characteristics (age, sex, ethnicity, [patient level] deprivation), lifestyle factors (smoking status [never, current, ex-smoker], alcohol consumption [units per week], and BMI), pre-treatment blood pressure readings (in the preceding 12 months), co-morbidities (rheumatoid arthritis, hypercholesterolemia [Read code for hypercholesterolemia, or most recent total cholesterol value >7.5mmol/l])”

7. Table 1: Proportions of patients exiting the study must be shown

Response

These are now given in the text on page 7, line 236-240, but since table 1 refers to methods, not results, we do not think it is appropriate to include this information in table 1 as well. The results section describes:

“Patients spent a median time of 2.6 years in the study (IQR 0.9, 5.5) and exited the cohort due to development of stage 2 hypertension (6,269 patients [5.8%]), a cardiovascular risk factors or score >20% (3,496 patients [5.0%]), advice being given (18,647 patients [17.1%]), treatment prescribed (35,305 patients [32.4%]), end of study follow-up (38,515 patients [35.4%]) or a combination of reasons (4,656 patients [4.3%]).”

8. Table 1: The authors state that patients with a CVD risk of >20% exited the study. This could be a contradiction since the authors state in their strengths and weaknesses that Low risk patients were defined according to individual risk factors, rather than an established risk calculator due to large amounts of missing data (e.g. for cholesterol, BMI), potentially underestimating the number of truly low risk patients available for analysis.

Response:

We thank the reviewer for pointing this out. It is true that the criteria for entry included high CVD risk based on the presence of CVD risk factors rather than CVD risk scores. Patients did exit the study if a high CVD risk score was coded in the notes during follow-up which happened for 1,955 individuals (1.8%) but we did not attempt to estimate or impute scores to define the population for the primary analysis. Since this analysis was pre-specified, we could not change the primary analysis, but our post-hoc sensitivity analyses excluding patients from entry into the cohort based on risk score did address this discrepancy and found no differences in the primary findings. We now draw attention to this better in the discussion:

“The focus of this study was on ‘uncomplicated’ low risk patients, but for pragmatic reasons, no attempt was made in the primary analysis to exclude those with a high risk of cardiovascular disease according to an established risk calculator due to large amounts of missing data (e.g. for cholesterol, BMI). Patients did exit the cohort if they had a coded high CVD risk score during follow-up. Whilst it is possible that some high risk patients may have been included in the study cohort initially, sensitivity analysis excluding patients with known high risk scores and those with a high estimated QRISK2 score did not affect the study results”

9. Table 2. The age thresholds of >55 years (men) and >60 years (women) are simple markers in hypertensive patients to identify those with at least moderate absolute total CVD risk. So, I would suggest adding the proportion of patients with an age over these thresholds to this table

Response

We have added these proportions to table 1.

VERSION 2 – REVIEW

REVIEWER	Alberto Morales Salinas Cardiocentro "Ernesto Che Guevara". Cuba
REVIEW RETURNED	05-May-2018
GENERAL COMMENTS	the authors included the main recommendations